# *Mucorales*/*Fusarium* Mixed Infection in Hematologic Patient with COVID-19 Complications: An Unfortunate Combination

**DOI:** 10.3390/pathogens12020304

**Published:** 2023-02-12

**Authors:** Andrea Marino, Maddalena Calvo, Laura Trovato, Guido Scalia, Maria Gussio, Ugo Consoli, Manuela Ceccarelli, Giuseppe Nunnari, Bruno Cacopardo

**Affiliations:** 1Department of Biomedical and Biotechnological Sciences, University of Catania, 95123 Catania, Italy; 2Unit of Infectious Diseases, Department of Clinical and Experimental Medicine, ARNAS Garibaldi Hospital, University of Catania, 95122 Catania, Italy; 3U.O.C. Laboratory Analysis Unit, A.O.U. “Policlinico-Vittorio Emanuele”, Via S. Sofia 78, 95123 Catania, Italy; 4Unità Operativa Complessa (UOC) di Ematologia, ARNAS Garibaldi Hospital, 95122 Catania, Italy; 5Unit of Infectious Diseases, Department of Clinical and Experimental Medicine, University of Messina, 98124 Messina, Italy

**Keywords:** invasive fungal infections, Fusarium infections, Fusarium/Mucorales infections, antifungal stewardship

## Abstract

Hematological diseases, especially those causing severe neutropenia, represent the main factor in the development of invasive fungal infections (IFIs). Furthermore, COVID-19 has been considerably associated with IFIs due to immunological dysregulation, prolonged hospitalization in intensive care units, and immunomodulatory therapies. Opportunistic molds are correlated with elevated morbidity and mortality rates in these patients, due to immune impairment, diagnostic complexity, and therapeutic challenges. Among opportunistic fungal infections, the *Mucorales* and *Fusarium* species are considered particularly aggressive, especially during severe neutropenia. A mixed *Mucorales*/*Fusarium* infection has been rarely described in scientific literature. Herein, we report a case of *Mucorales* and *Fusarium* co-infection in a patient with acute leukemia whose clinical history was also complicated by COVID-19. Herein, we report a challenging case in order to encourage the clinical suspicion of combined fungal infections in immunosuppressed patients, performing a punctual microbiological diagnosis, and promptly administering the correct empiric and targeted antifungal therapy.

## 1. Introduction

Hematological diseases play a key role as predisposing factors for invasive fungal infections (IFIs). Clinical complications such as neutropenia, treatment-related immunosuppression, and mucosal debility significantly increase the odds of developing IFIs, especially during prolonged hospitalization [1]. Furthermore, COVID-19 has been strongly associated with IFIs due to immunological dysregulation, an extended length of stay in intensive care units, and immunomodulatory therapies (corticosteroids or others) [2]. Opportunistic molds correlate with elevated infection and mortality rates in these patients. Specifically, the *Aspergillus* species is known to express high virulence characteristics and angioinvasion ability, taking advantages of hematologic patients’ critical clinical condition. Among fungal pathogens, the *Mucorales* and *Fusarium* species are also considered aggressive during severe immunosuppression. A mixed *Mucorales*/*Fusarium* infection has been rarely described [3]. Herein, we report a case of *Mucorales* and *Fusarium* co-infection in a hematologic patient whose clinical history was also complicated by COVID-19 pneumonia. Although the outcome was unfortunate, our case highlights the importance of promptly suspecting and diagnosing possible mixed infections in critical patients.

## 2. Case Presentation

An 83-year-old male was admitted to the Emergency Department due to fever (up to 38.5 °C), cough, severe asthenia, and the appearance of an ulcerative lesion on the nose. The patient reported a story of losing around 5 kg in the previous 30 days. On admission, the nasopharyngeal swab tested positive for SARS-CoV-2 and the patient was transferred to the Infectious Diseases Unit. His medical history showed gastroesophageal reflux disease, paroxysmal supraventricular tachycardia, and obliterans chronic arteriopathy. He took betablockers, cilostazol, and proton pump inhibitors. The patient was fully vaccinated (three doses) against SARS-CoV-2. On admission, he was febrile (T: 38 °C), blood pressure was 130/80 mmHg, heart rate was 70 bpm, oxygen saturation in room air was 91%, and respiratory rate was 18 breaths/min. Glasgow Coma Scale (GCS) was 14. Arterial blood gas analysis showed hypoxemia and a Venturi Mask 10 lt/FiO2 40% was positioned. Physical examination displayed decreased vesicular breath sounds in the right thorax and an ulcerative lesion on the left side of the nose (Figure 1), which were analyzed by biopsy. A chest X-ray and subsequent thorax CT scan highlighted right medio-basal lung consolidation, perivascular interstitial thickening, and bilateral pleural effusion (Figure 2). In addition, diffuse mediastinic lymphadenopathy and ascending aorta aneurysm (4.8 cm) was highlighted. The abdomen and sinus CT scans did not show abnormal findings.

Blood tests revealed pancytopenia with a low white blood cell count (WBC 0.9 × 10^3^/mm^3^, Neutrophils 1.6%, Lymphocytes 56.2%, Monocytes 42%), anemia (Hb 8.76 g/dL), and a low platelets count (41 × 10^3^/mm^3^). Inflammatory markers were elevated: C-Reactive Protein (CRP) was 26.42 mg/dL (normal range < 0.5 mg/dL), Erythrocyte sedimentation rate (ESR) was 97 mm/h (normal range < 10 mm/h). Procalcitonin was 2.6 mcg/L (normal value < 0.1 mcg/L). Transaminases and bilirubin levels were normal, as well as coagulation parameters. Creatinine was 0.9 mg/dL with an eGFR of 79 mL/min. The patient also presented with hypoalbuminemia (2.24 g/dL) and high ferritin levels (2000 ng/mL). Glucose levels were normal. HIV, HBV, and HCV serology tested negative. Legionella and pneumococcal urinary antigens also tested negative. Blood cultures and urine cultures tested negative, as well as serum EBV- and CMV-DNA. Serum beta-glucan was 294 pg/mL (normal values < 60 pg/mL), and serum galactomannan was negative. Empiric antibiotic therapy was started with intravenous (IV) piperacillin/tazobactam 4.5 gr three times daily and teicoplanin 400 mg daily (after loading dose); empirical antifungal therapy was administered with IV liposomal amphotericin B at a dosage of 5 mg/kg. In addition, supportive COVID-19 therapy was started with corticosteroids. A biopsy and a bone marrow aspirate were performed, and acute myeloid leukemia was diagnosed. In addition, microscopic examination of the ulcerated nasal lesion revealed the presence of septate and non-septate hyaline hyphae (Figure 3 and Figure 4). Septate hyphae were attributed to *Fusarium (F.) solanii*, whose colonies were revealed after culture exams (Figure 5). Non-septate hyphae were suggestive of possible zygomycete infections. Considering the low culture sensitivity in the case of zygomycosis, a molecular assay was performed to further investigate the microscopic records. The multiplex real-time PCR assay (MucorGenius^®^, PathoNostics, Maastricht, The Netherlands), which targets the *Mucorales* 18S rDNA, showed a positive result. Antimicrobial susceptibility testing was provided only for the *F. solanii* strain by using broth microdilution (SensititreYeastOne^®^ method; Thermo Fisher Scientific, Cleveland, OH, USA). The test revealed a significant increase in MIC values for all of the tested antifungal drugs. Specifically, the following MIC values were recorded: >8 mg/L for anidulafungin, caspofungin, micafungin, amphotericin, posaconazole and voriconazole; >256 mg/L for fluconazole; >64 mg/L for flucytosine; >16 mg/L for itraconazole. Intravenous antibiotic therapy was administered for 18 days, obtaining a moderate decrease in inflammatory marker levels along with pulmonary imaging improvement. However, clinical conditions continued to deteriorate due to leukemia. In addition, serum Beta-D-glucan levels were decreased. According to microbiological reports, liposomal amphotericin B was maintained at the same dose combined with oral posaconazole 300 mg/day (with previous loading dose). Prophylactic cotrimoxazole was added. Furthermore, corticosteroids, as suggested by hematologists, were maintained. After 18 days, SARS-CoV-2 tested negative on a nasopharyngeal swab and the patient was transferred to the Hematology Unit, where he died after 20 days due to cardiac arrest.

## 3. Discussion

IFIs are associated with high morbidity and mortality in patients with hematological malignancies, especially for those affected by acute leukemias, with prolonged neutropenia being a major risk factor [3].

Here, we described a patient suffering from acute myeloid leukemia who developed a mixed fungal infection from *Mucorales*/*Fusarium*, both recovered from skin biopsy. Furthermore, clinical conditions were compromised by SARS-CoV-2 infection and concomitant pneumoniae impacting a dysregulated immune system.

Among *Fusarium* species, *F. solanii* is the most frequent microorganism associated with human diseases, followed by *F. oxysporum* and *F. verticillioides* [4]. Those fungi, ubiquitous in the environment, are considered serious opportunistic pathogens for immunocompromised patients with a mortality rate of up to 70% [5].

Although the airways represent *Fusarium’s* primary way of entry by inhalation of airborne conidia, leading to severe sinusitis or pneumoniae, skin is the second most frequent entrance, resulting in localized or disseminated clinical forms, especially among immunocompromised patients [4]. As discussed by Nucci et al. [6], among neutropenic subjects, both localized and disseminated *Fusarium* skin localizations are characterized by high mortality rates. Different cutaneous manifestations of *Fusarium* have been described, mostly depending on the host immune status, varying from disseminated disease with painful nodular or papular ecthyma-like lesions at different stages of evolution, to localized target forms developing in any site with rapid evolution to necrotic forms.

The management of *Fusarium* infections may be significantly challenging, due to antifungal resistance determined by drug efflux mechanisms, biofilm formation, and target alterations [7]. Azole intrinsic resistance is a major concern, mainly caused by the wide use of azole in plant protection [8]. Furthermore, echinocandins show less efficiency against *Fusarium* species, complicating patients’ therapeutic management [9].

Regarding the treatment of cutaneous fusariosis, guidelines suggest surgical debridement along with antifungal therapy. Although different species could be susceptible to different drugs, liposomal amphotericin B is the preferred option. Studies have reported combination options with azoles (voriconazole or posaconazole), with no conclusive results [4]. Undoubtedly, therapy response strictly correlates with hematological disease resolution and/or neutrophil recovery [10].

On the other hand, cutaneous mucormycosis is the third most typical clinical manifestation of *Mucorales* infections, following rhino-cerebral and pulmonary forms. Skin involvement could be localized, extended (muscles, bones), or disseminated to noncontiguous organs [11]. Imitating *Fusarium*, *Mucorales* is ubiquitous in the environment and its spores could be recovered in compost piles, soil, fruits, and decaying vegetation. Hematological conditions along with immunosuppressive diseases represent key risk factors for the development of mucormycosis [12,13].

Cutaneous mucormycosis usually develops as a single, indurated, painful area of cellulitis evolving in ecthyma-like lesions. Dissemination and deep tissue involvement are uncommon complications. Intravenous liposomal amphotericin B is the drug of choice to treat *Mucorales*, whereas posaconazole and voriconazole could be used as a salvage therapy for those patients who do not tolerate amphotericin [14]. According to current guidelines, mucormycosis diagnostic assessments should be performed by microscopic examination and fungal culture on tissue biopsy (in case of suspected cutaneous forms) or sinus washing in case of rhinocerebral disease [15]. Unluckily, these conventional methods suffer from low sensitivity due to the intrinsic fragility of zygomycetes coenocytic hyphae. Real-time PCR assays have been recently validated for zygomycetes detection in clinical samples such as tissue biopsies, bronchoalveolar-lavage fluids, and serum samples. These molecular technologies offer the possibility to improve sensitivity yield, providing fast results for severe and time-dependent infections. *Mucorales* species involved in human infections can be correctly detected through easy-to-use PCR assays, which should be integrated into all diagnostic workflows performed for mucormycosis clinical suspicion [14,16]. *Mucorales* do not share 1,3-Beta-D-glucan and galactomannan as cell wall components and these tests are negative in patients with mucormycosis.

Regarding *Fusarium* spp., although more studies are needed, Beta-D-glucan could be helpful to rule out fusariosis, due to its high negative predictive value (99%). However, it should not be used to decide treatment initiation due to a very low positive predictive value (7%) and low specificity (54%) [17].

A Galactomannan assay performed to diagnose Aspergillosis could cross-react in patients with *Fusarium* infections. Neither the number of positive tests nor galactomannan value is useful to discriminate between invasive aspergillosis and invasive fusariosis, especially in regions where fusariosis is uncommon [18]. Although not statistically significant, the patient we described had positive serum beta-D-glucan and a negative value for serum galactomannan.

Several authors remarked on the importance of integrating rapid polymerase chain reaction-based methods into fungal infection laboratory diagnosis [19,20,21]. According to these literary data, in critically ill patients, fungal DNA detection is a promising method to rapidly detect pathogens. Sensitivity rates can reach 43 to 100%, while specificities ranges are approximately 64 to 100%. Fluctuations are justifiable depending on the microorganisms count or clinical sample interferences. Otherwise, the sensitivity percentages of culture-based methods reach no more than 25–50% [16,21].

Among bacterial and viral superinfections [22,23,24], COVID-19 has been widely associated with mucormycosis through several pathological patterns such as immune system impairment due to cytokine storm [25], impaired phagocytosis, and endothelitis [26,27]. In addition, immunosuppressive therapies such as corticosteroids and immunomodulatory therapies such as anti-IL6 antibodies favor the development of COVID-19 associated mucormycosis (CAM) [28,29,30,31].

The patient we described had acute leukemia along with COVID-19 pneumonia. In addition to the two major risk factors for fungal diseases such as severe neutropenia and SARS-CoV-2 infections, the patient was also aged and hospitalized.

Based on worsening clinical conditions and laboratory results, empirical antifungal therapy was started before the results of the skin biopsy examination. Due to the poor general status of the patient, surgical debridement of the skin lesions was not performed. Bronchoscopy with bronchoalveolar lavage was also not performed, as it was challenging to fulfill because of the patient’s severe respiratory conditions. A *Fusarium* resistant pattern would not have allowed any therapeutical choice, although liposomal amphotericin B was confirmed and posaconazole was added; the same combination could have been appropriate for *Mucorales* infection, as reported by guidelines. Although pharmacologic therapies were used, insufficient immune recovery due to the rapid progression of leukemia contributed to the patient’s unfortunate outcome.

There are few reports in scientific literature about *Mucorales*/*Fusarium* co-infection, and little data about its clinical management and therapeutic approach. Based on treatment guidelines about these infections alone, our clinical point of view about co-infection suggests that liposomal amphotericin B may represent a valid empirical choice, due to its wide fungal coverage and safety. However, both novel and classic diagnostic methods, along with precise drug susceptibility tests, should be performed as soon as possible. The rationale should be to promptly provide the correct diagnosis, avoiding clinical and therapeutical delays.

## 4. Conclusions

Among the general population, the incidence of IFIs is gradually rising. This trend could be explained by both the increasing prevalence of immunocompromised patients and the significative improvement in pathogenic fungi diagnosis. *Mucorales* and *Fusarium* infections represent both an example of difficult to diagnose pathogens and a paradigm of difficult-to treat germs, becoming challenging clinical conditions to deal with [32], especially considering mixed infections, which are uncommon, especially in our region. Microscopic and culture-based investigations still represent the gold standard in fungal diseases work-up, allowing for definitive diagnosis and an antifungal susceptibility test. The limitations of these methods, such as the time-to-result parameter, could be fixed by more sensitive and faster molecular techniques which should be integrated into fungal infection laboratory diagnosis [19,20,21,33]. As a final consideration, the rapid progression of severe fungal infections should be faced through both a prompt clinical suspicion and recognition along with a timely diagnostic protocol. These diagnostic tips could play a key role in assuring optimal clinical and therapeutical patient management.

## Figures and Tables

**Figure 1 pathogens-12-00304-f001:**
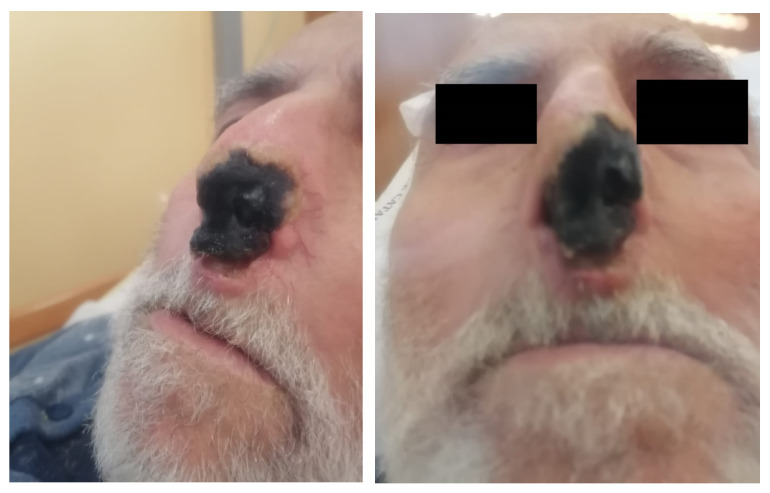
Echtyma-like necrotic-ulcerative lesions.

**Figure 2 pathogens-12-00304-f002:**
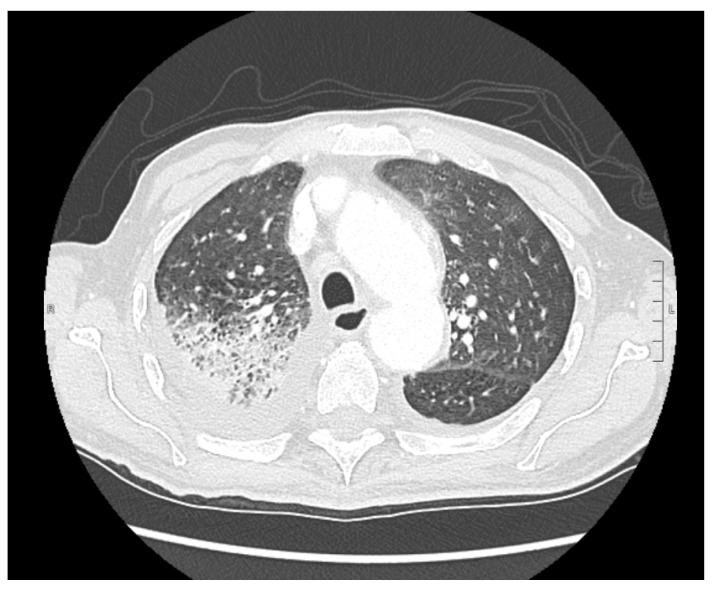
Thorax CT scan showing right medio basal consolidation, pleural effusion, and ascending aorta aneurysm.

**Figure 3 pathogens-12-00304-f003:**
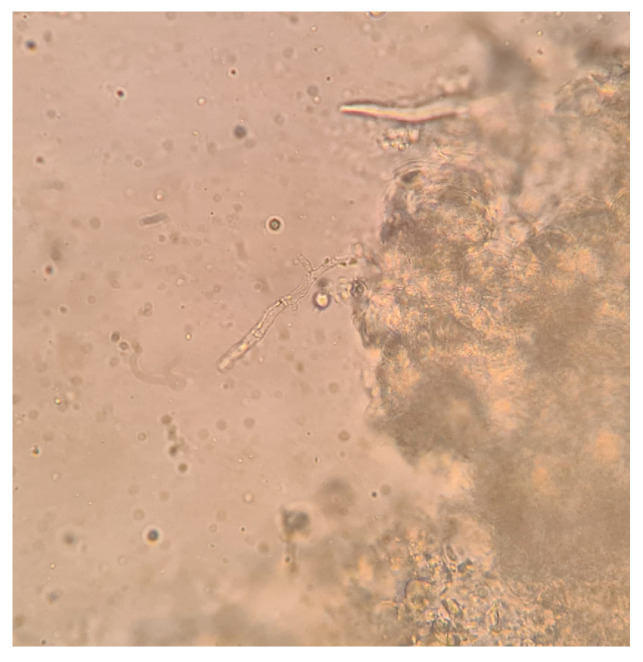
Septate hyphae from microscopic investigation on biopsy samples (40× magnification).

**Figure 4 pathogens-12-00304-f004:**
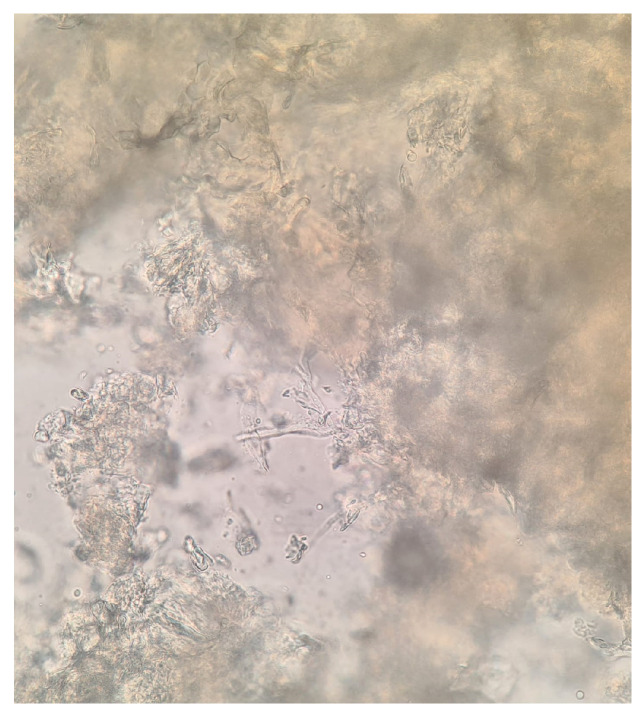
Non-septate hyphae from microscopic investigations on biopsy samples (40× magnification).

**Figure 5 pathogens-12-00304-f005:**
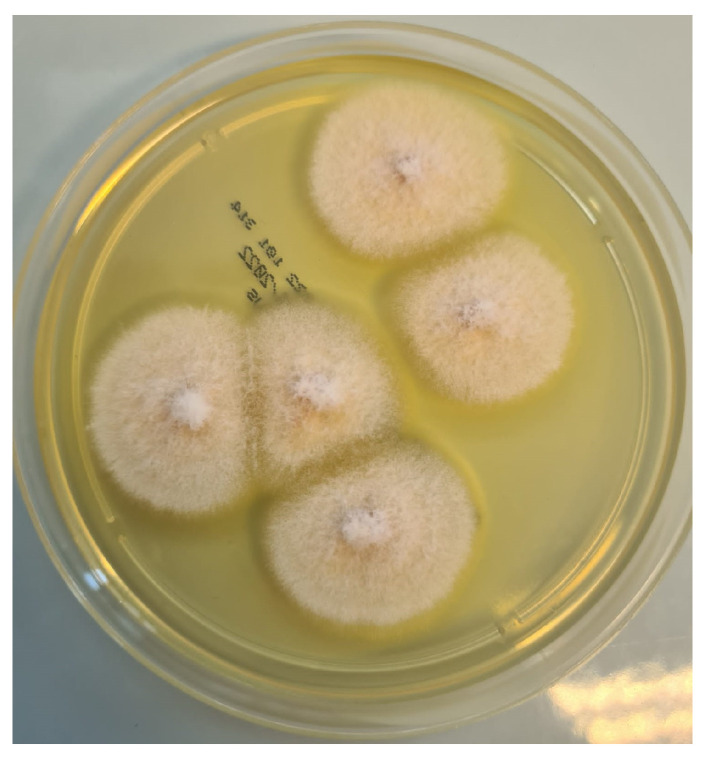
*Fusarium solanii* colonies on Sabouraud culture examination on biopsy samples.

## Data Availability

Not applicable.

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
