# Peer review of "Mucorales/Fusarium Mixed Infection in Hematologic Patient with COVID-19 Complications: An Unfortunate Combination"

_pathogens, 2023, doi:10.3390/pathogens12020304_

Round 1

Reviewer 1 Report (Previous Reviewer 3)

I appreciate the time and consideration the authors have taken to improve their manuscript. Although the English has been edited, the readability of this manuscript is still challenging with often inappropriate use or choice of words. For example, in the abstract, the line "Our cumbersome clinical case, which ended with an inauspicious outcome aims to rise the clinical suspicion of uncommon fungal infections in immunosuppressed patients, to perform a punctual microbiological diagnosis to detect uncommon fungal infections, and to prompt administer the correct empiric and target antifungal therapy" does not make sense with the adjectives used.

As the authors discuss, invasive fungal infections are well described in both patients with hematologic malignancies and those with severe COVID-19. If this case is accepted for publication, the majority of the discussion should revolve around how the management of co-infection of these two pathogens differs from the priniciples of managing these infections alone as published in other studies and guidelines.

Author Response

Thank you for your precious suggestions. We rephrased the abstract and check the English language of the whole manuscript. As regards your suggestions about discussion section, we did not find any data about the treatment of Mucorales/Fusarium co-infection, as well as there are not guidelines about contemporary treatment. This is another reason why we believe in the novelty of our case. To fulfill what you suggested, we added some lines about our clinical point of view at the end of the discussion section. We hope you would appreciate.

Reviewer 2 Report (Previous Reviewer 2)

Hi, 

I congratulate the authors for the significant case study. However, I would still suggest a fine revision of the english language only to further improve the readability. 

Also please mention the full forms first before introducing the short forms even in the abstract (example, IFI). 

Please discuss, If there is any information about the ketoacidosis or diabetic status of the patient. 

Thanks and warm regards

Author Response

Reply: Thank you for your precious suggestions. We rephrased the abstract and check the English language of the whole manuscript. As regards clinical details, the patient hadn’t diabetes. His risk factor was the neutropenia due to severe leukemia.

Reviewer 3 Report (Previous Reviewer 1)

In the manuscript titled “Mucorales/Fusarium mixed infection in hematologic patient 2 complicated by COVID-19: an unfortunate combination” present a case of a mixed fungal infection Mucorales/Fusarium in a patient with acute myeloid leukemia complicated with COVID-19 and the manuscript quality was improved with the revision.

Author Response

Thank you for your kind comment.

Reviewer 4 Report (New Reviewer)

Please have a look "The multiplex real-time PCR 106 assay (MucorGenius®, PathoNostics, Maastricht, The Netherlands), which targets the  Mucorales 18S rDNA, showed a positive result. Antimicrobial susceptibility testing was provided only for F. solanii strain by using broth microdilution (SensititreYeastOne®  method; Thermo Fisher Scientific, Cleveland, OH, USA)."

Why author pefered the broth microdiution for F. Solanii over PCR/Sequencing? 

Normand, A. C., Imbert, S., Brun, S., Al-Hatmi, A. M., Chryssanthou, E., Cassaing, S., ... & Fekkar, A. (2021). Clinical origin and species distribution of Fusarium spp. isolates identified by molecular sequencing and mass spectrometry: a European multicenter hospital prospective study. Journal of Fungi7(4), 246.

Author Response

Thank you for the observation. Our fusariosis diagnostic workflow included broth microdilution as a susceptibility testing assay, while molecular techniques are useful to perform an identification. In our setting, a commercial and easy-to-use molecular assay for Fusarium is not available. Furthermore, sequencing it is not an option in a routinary diagnostic laboratory. In conclusion, we reserve future studies to experimental investigate about our local Fusarium spp. epidemiology through sequencing. Finally, we appreciate the kind suggestion about the interesting reference, which we already cited in our manuscript.

This manuscript is a resubmission of an earlier submission. The following is a list of the peer review reports and author responses from that submission.

Round 1

Reviewer 1 Report

In this case report, authors report a case of a mixed fungal infection Mucorales/Fusarium in a patient with acute myeloid leukemia complicated with COVID-19, and discuss clinical manifestation, diagnosis and therapy of Fusarium infections and cutaneous mucormycosis. This case report has some problems as following:

1.     Please make the abbreviations explicit as ED (Pag 2 – line 53), MRGE (Pag 2 – line 57), GCS (Pag 2 – line 62).

2.     Whether the patient underwent sinus CT examination? Can you provide the results and images of sinus CT?

3.     Can you provide Chest CT images?

4.     Has the febrile patient received blood culture examination ? Why not ?

Reviewer 2 Report

Dear authors, 

First, I would like to congratulate you for your work. It is a very interesting case of fungal co-infection in a debilitated patient. However, I do have a few concerns regarding the report which I hope will further aid in enhancing the readability of the report. 

1. Overall language of the paper needs improvement not majorly from the scientific but from the fluidic perspective. For the readability. (Example, the case report is about a hematological malignancy patient developing fungal co-infection. However, it has been stated in the manuscript that he was diagnosed with AML (line 86); there are few words if substituted I guess would make it sound better (example Line 35: primate; Line 48: inauspicious etc.)

2. Abbreviated forms have been used before predefining them. Please avoid doing so even for the obvious or the frequently used ones (line 53: ED, line 72: Ne, Mo, Ly etc.).

3. [Line 44: among these pathogens] This line needs to be re-written as before that has been mention of Aspergillus not the mycelial fungal infections.

4. The patient had developed mucorales so it will be nice to add if checked for diabetic ketoacidosis (found or not found)

5. Add magnification details to the microscopy pictures. 

6. The microbroth picture does not add anything neither does the Fusarium solani also as it doesnot show its characteristic color. Also preferably it should be cultured in tubes to avoid risk of any contamination.

7. Please remove the unnecessary test details which don't add anything. Also it was clear that the treatment of choice will be Amphotericin B for these two fungi. So, discussing treatment as a question in discussion is not important. Can discuss about the serological markers by mentioning about the false positive possibility of galactomannan with Fusarium species, if people get it in their labs. Because every lab can not afford molecular diagnosis. 

8. Shorten the discussion and conclusion. Make it super crisp. Also the conclusion should not exceed 1 paragraph. 

9. Cite some more cases to give a broader perspective about these infections in hematological malignancy patients. 

Reviewer 3 Report

Thank you for giving me the opportunity to review this case. While this is an interesting, but unfortunate, case, invasive mold infections in leukemic patients are well described, including mixed infections. More details in the discussion would be pertinent to highlight why this case is unique or what it adds to medical literature as a whole.